# Algorithm to Generate Trajectories in a Robotic Arm Using an LCD Touch Screen to Help Physically Disabled People

Yadira Quiñonez [1], Jezreel Mejía [2], Oscar Zatarain [1,*], Carmen Lizarraga [1,*], Juan Peraza [1] and Rogelio Estrada [1]

[1] Facultad de Informática Mazatlán, Universidad Autónoma de Sinaloa, Culiacán Rosales 80020, Mexico; yadiraqui@uas.edu.mx (Y.Q.); jfperaza@uas.edu.mx (J.P.); restrada@uas.edu.mx (R.E.)
[2] Centro de Investigación en Matemáticas, Zacatecas 98160, Mexico; jmejia@cimat.mx
[*] Correspondence: zatarainpedrero@gmail.com (O.Z.); carmen.lizarraga@uas.edu.mx (C.L.)

**Abstract:** In the last two-decade, robotics has attracted a lot of attention from the biomedical sectors, to help physically disabled people in their quotidian lives. Therefore, the research of robotics applied in the control of an anthropomorphic robotic arm to people assistance and rehabilitation has increased considerably. In this context, robotic control is one of the most important problems and is considered the main part of trajectory planning and motion control. The main solution for robotic control is inverse-kinematics, because it provides the angles of robotic arm joints. However, there are disadvantages in the algorithms presented by several authors because the trajectory calculation needs an optimization process which implies more calculations to generate an optimized trajectory. Moreover, the solutions presented by the authors implied devices where the people are dependent or require help from other people to control these devices. This article proposes an algorithm to calculate an accuracy trajectory in any time of interest using an LCD touch screen to calculate the inverse-kinematics and get the end-point of the gripper; the trajectory is calculated using a novel distribution function proposed which makes an easy way to get fast results to the trajectory planning. The obtained results show improvements to generate a safe and fast trajectory of an anthropomorphic robotic arm using an LCD touch screen allowed calculating short trajectories with minimal fingers moves.

**Keywords:** distribution function; linear transformation; trajectory planning; anthropomorphic robotic arm; LCD touch screen; arduino mega; MATLAB

## 1. Introduction

In the last two decades, automation and control have become a topic of interest for researchers from different areas. Mainly in industrial robotics [1,2] and in the medical area's robotic systems. There is a wide variety of automation and control; however, the medical area has been greatly benefited; below, some works related to this area are mentioned, such as tele-operated surgery [3], surgical pattern cutting [4]. According to Shademan et al., an industrial robot's precision offers excellent advantages in this area [5]. The trajectory of a robotic arm has been a subject widely studied by different researchers using different techniques. These techniques help us to understand and make different tasks; for example, in the industries, a robot arm makes repeated work where it is making a trajectory. This topic has been improved throughout time, and nowadays, some related works with robotic assistance arms have been developed [6–8] to help people with physical disabilities to become more independent in daily life. According to the works in [9,10], the most commonly used methods to calculate trajectories in robotic arms are cubic polynomials [11], trapezoidal trajectory [12], and the Euler angles [10]. With these methods, it is possible to obtain smooth motions in the joint space, and these movements are predictable in the task space.

However, these methods present specific problems when calculating the trajectory. According to the work of Sidobre and Desormeaux [11], with cubic polynomials, it is

difficult to calculate the trajectory in an optimal time; that is, there is no freedom to take a trajectory with a time of 1 s or another to ensure accurate results in the trajectory. Hence, to obtain an accurate trajectory, it is necessary to estimate the optimal time, and therefore, more calculations are required. According to [13], the (N-1)-order polynomials present other disadvantages when generating the trajectories. First, it is only possible to generate velocity and acceleration for a determinate position, and it is not possible to change those positions. Second, when the polynomial order increases, the robotic arm acquires an oscillatory behavior, which causes an unnatural response, and therefore, the precision of the trajectory decreases. Third, the constraint equations system is difficult to solve, and compile-time is increased as a result. Finally, the polynomial coefficients have to be recomputed every time that a new endpoint is assigned.

In relation to the trapezoidal trajectory, there are some disadvantages. According to [12], although it does not require many computations, this trapezoidal profile requires additional response-time and has a constant acceleration, velocity, and deceleration regions, which means, large jerk causes vibrations which restrict position accuracy. Finally, another method that is very popular to calculate the trajectory of a robotic arm is Euler angles. According to [10] which can cause malfunction of robots with systematic errors and lead to undesirable results because of the nonlinearity of the three Euler angles.

In addition to the disadvantages presented in each of the methods, it is essential to mention that when estimating the velocity and the acceleration in a real application, these calculations can be affected by noise during the process. As an alternative to this problem, some authors [14–16] have proposed different approaches to reducing possible measurement errors and noise. In the work of Mercorelli [14] a velocity estimator based on current measurements is proposed. In another work [15] a derived approximation structure is proposed to estimate velocity by measuring position. In [16] the authors have proposed an adaptive algorithm to calculate a signal's derivative with convergence in infinite time.

Some work has proposed to use a touch screen for robotic control as an alternative solution to help people with physical disabilities, providing an intuitive interface. Makwana and Tandon [17] have proposed controlling a robotic wheelchair using a touch screen to improve a person's movements with physical disabilities. Similarly, Bularka et al. [18] present an alternative to control a robotic arm using the accelerometer of a watch and a smartphone; control is carried out using finger gestures on the touch screen or moving the phone in the air. The authors in [19] present solutions for robotic control using the potential advantages of smartphones so that people with a physical disability can interact with the environment in a friendly way.

Another problem related to robotic arms control is when a joystick or another kind of sensor is used. According to [20], some inconveniences can be presented, for example, it has been proved that controlling a robotic arm by voice commands lack high reliability, or when using keyboards, joystick or model controls to generate a robotic arm trajectory. However, these techniques are unsatisfactory because disabled persons cannot provide inputs to keyboard and joysticks for a sustained period.

To solve all problems mentioned regarding arm control and the calculation of their trajectory, first of all, it has been considered an LCD touch screen to generate the endpoint motion of the robot because the control is intuitive, the people are more familiarized using an LCD touch screen due to they use smartphones, laptops, etc.; which count with an LCD touch screen to work. Also, the idea to use an LCD touch screen to control an anthropomorphic robotic arm with just one finger and using the distribution function proposed, it is a great option to help physically disabled people in their quotidian lives with an exact, fast and comfortable option to control the robotic arm. Concerning the problems mentioned above when calculating a robotic arm's trajectory, it is proposed to use a distribution function, named distribution function-($\zeta at_s$), that allows the calculation of an exact trajectory and for any parameter. For example, the velocity and acceleration can be chosen without affecting the desired position and the initial position by changing one parameter and making one calculation to generate the trajectory planning.

The remainder of this paper is organized as follows: Section 2 introduces an overview of previous related works on the application of different techniques to generate the trajectory of a robotic arm. Section 3 describes how the LCD touch screen works in order to understand the algorithm operation. Section 4 presents the proposed algorithm to generate a safe and fast trajectory; it includes the conditions, the linear transformation as baseline to calculate the inverse-kinematic with the distribution function and its properties to generate de trajectory and finally how is calculated the velocity and acceleration. Section 5, simulation and results, and designs land on the code using the Arduino Mega microcontroller to obtain the end-point of the robotic arm and MATLAB to generate the trajectory. Section 6 mentions a critical discussion of the results obtained with the proposed algorithm compared to the algorithm of cubic polynomials. Finally, Section 7 summarizes the conclusions of the paper and indicates further work.

## 2. Related Works

Most robotic applications are currently related to industrial robot arms; therefore, it is essential to know the main concepts in robotics, such as kinematics, dynamics, movement planning, computer vision, and control. In this sense, robotic control is considered the core part when planning trajectories and the most used technique to achieve a specific position is inverse-kinematics.

Liu et al. [21] analyzed the Denavit-Hartenberg (DH) method for planning trajectories with multiple manipulators; besides, the authors solved the inverse kinematics equation employing the matrix operation algebraic method. In another work [22], the authors proposed the exponential product method based on screw theory to avoid the singularity problem. They used different analytical, geometric, and algebraic methods combined with the Paden-Kahan subproblem and the matrix theory. Other works [23–27] proposed conventional tools to discover the robot manipulator's kinematic solutions.

Other methods have been used to control a robotic arm; for example, the authors [28] designed and programmed an interface with open control algorithms to generate and control three types of movements: joint interpolation, linear interpolation, and circular arcs. The results have been tested on the CompactRIO industrial controller. According to the results obtained, some position errors were found in the end effector, and the process shows oscillations on the desired coordinates. In the work of Barghi-Jond et al. [29] proposed the use of 3rd, 4th and 5th-degree polynomial trajectory to present three problems of optimization of trajectories, and it has been mentioned that the slowest trajectory occurs when there is a high-degree polynomial. The coefficients of the polynomial cannot get frequently.

In other work [30], the authors have used artificial intelligence to optimize trajectory planning; mainly, this work is focused on an improved genetic algorithm. The authors mention that the polynomial interpolation method is an excellent option to study robot obstacle avoidance due to the complexity of this method. Then, the trajectory of the robot's joints is planned by the quantic polynomial; however, the optimal time of this method is 13.729 s because a smooth motion is presented at this time, and the trajectory planning presents oscillations. Other authors [31] used trapezoidal motion to generate trajectories; nevertheless, it is necessary to reduce the residual vibration in acceleration and deceleration. According to [12], the trapezoidal trajectory has a constant velocity and acceleration, and a large jerk causes vibration, the accuracy of the results decreases and requires an additional response time.

In another research work [32], the authors have proposed optimizing the trajectory planning using the Gaussian distribution.

The research of robotics applied in the area of assistance and rehabilitation has increased considerably, kinds of research works have focused on motion planning using manipulators with different degrees of freedom to allow the independence of patients who require rehabilitation or assistance in daily life activities [33,34]. Most of the research related to the trajectory generation for assistance or rehabilitation manipulators are based on the use of different techniques such as: forward and inverse-kinematic analysis [35,36], DH method [37], the Jacobian Equation [38], the geodetic curves [39], and the minimum jerk algorithm [40], among others. All methods are used in order to improve the safety and the experience of patients who need to perform human-robot interaction to recover the functional impairment or to help the functional rehabilitation of people with loss of autonomy [41–43].

As a result of the analysis of the related works, it can be highlighted there are several research works that present solutions both Robotic controls, as well as robotics, applied in the area of assistance and rehabilitation. However, several disadvantages in these algorithms are presented, the trajectory can oscillate between feasible and infeasible during optimization, and it needs an optimization process which implies more calculations to generate an optimized trajectory. Moreover, the solutions presented by the authors were applied with joystick MATLAB, Lab-VIEW, industrial controller CompactRIO and camera. This implies that people are dependent or require help from other people to control these devices. In addition, the trajectory captured in these devices is slower and more complex to generate the trajectory of a robotic arm, and the short trajectories cannot be calculated.

## 3. LCD Touch Screen Operation

Nowadays, using joystick is widespread and other types of electronics devices to control a robotic arm, whatever is the kind of robotic arm; cartesian, SCARA, cylindrical, delta, spherical, and anthropomorphic. Nevertheless, these sensors that are used as control tools could have problems when it is implemented within particular characteristics, and it cannot meet the necessities or expectations to get the maximum utilization. These problems are due to constant effort requirement and waiting time while a force is being applied until the robotic arm gets the desired position, without mentioning that it is needed to coordinate. Supposing that the patient is sick of diabetics (in the case of hypoglycemia), lupus, or another disease that impedes good coordination, so in this case, joystick could be a difficult tool used by patients who present these cases. Therefore, it has been thought in those cases and has been implemented the LCD touch screen rather than joystick or another device. Before to start explaining about the algorithm to obtain the trajectory of the anthropomorphic robotic arm using the LCD touch screen as a control.

First, how the LCD touch screen works, it has to be explained. Arduino Mega was utilized to program the interface and get the data to obtain the end-position of the robotic arm; it was used the libraries Adafruit_TFTLCD.h, Adafruit_GFX.h and TouchScreen.h. Then, the interface was programmed to reference the task space such as it can be observed in Figure 1a, so, four right triangles with equal areas, each time that the LCD touch was touched, it gives us a coordinate between 123 and 932 in the axis $X$ and, 897 and 128 in the axis $Y$. This coordinate is processed using MATLAB to obtain a coordinate $(x_p, y_p)$ where $x_p \in (-1, 1)$ and $y_p \in (0, 1)$. So, every coordinate which is less than 527.5 and greater or equal to 123, that means $-1 \leq x_p \leq 0$ and, if a coordinate greater than 527.5 is given and less or equal to 932, then $0 < x_p \leq 1$. this is the same for $y_p$; if a coordinate between 128 and 512.5 is given, then $0 \leq y_p \leq 0.5$, and if a coordinate between 512.5 and 897 is given, then $0.5 \leq y_p \leq 1$. Note that 527.5 is the middle point in the axis $X$ of the LCD touch screen and 512.5 is the middle point of the axis $Y$ of the LCD touch screen. Taking the coordinates $(x_p, y_p)$, it can be started to work with the algorithm presented in the next section.

## 4. Algorithm Description

The proposed algorithm has been thought for the anthropomorphic robotic arm, which has three rotation joints, and it has been inspired in third-order polynomial time scaling to propose the distribution function, which is discussed through this paper. In the next section, it is going to talk more about this distribution function. First, it is necessary to speak of the first conditions, second establish the linear transformation as baseline to calculate the inverse-kinematic with the distribution function and its properties to generate de trajectory and finally calculating the velocity and acceleration.

### 4.1. Nomenclature

$x_p$: coordinate in the axis X obtained from the LCD Touch screen.
$y_p$ : coordinate in the axis Y obtained from the LCD Touch screen.
$\alpha_p$ : angle to obtain the direction of the coordinate obtained from the LCD Touch screen.
$\alpha$ : total angle rotation of the first joint of the robotic arm.
$S_f$ : final position.
$q_0$ : initial position.
$t_s$ : final time of the trajectory.
$a$: parameter to change the shape of the trajectory, velocity, acceleration, and Jerk.
$n$ : constant greater to 1.
$\zeta$: parameter used to generate the trajectory through the time.

### 4.2. Description of Conditions

The joint which is more nearly to the base of the robotic arm has to rotate greater or equal to 0 degrees and equal or less to 180 degrees, and that would be represented $0 \leq \alpha \leq 180$, where $\alpha$ represents the first-joint degrees with a clockwise or counterclockwise rotation, depending on the point that has been touched on the LCD touch screen. Then, it can be obtained the coordinate $(x_p, y_p)$ which gives another datum to obtain $\alpha_p$ that represent the degrees of the condition that $(x_p, y_p)$ is positioned which is going to be calculated as follow:

$$\alpha_p = \begin{cases} \tan^{-1}\left(\frac{y_p}{x_p}\right) \\ \frac{\pi}{2} \ if \ x_p = 0 \ and \ y_p > 0 \\ 0 \ if \ x_p = y_p = 0 \end{cases} \tag{1}$$

where $x_p$ must be strictly differently than zero in the first part of (1) and for when $x_p = 0$ then $\alpha = \frac{\pi}{2}$. $x_p, y_p$ are represented like intervals, $y_p$ is contained in $(0, 1)$, $\alpha$ is contained in $(0, \pi)$ ($(0, 180)$ degrees) and $x_p$ is contained in $(-1, 1)$ and for when $\frac{\pi}{2} \leq \alpha \leq \pi$ happens, the gripper is located from the negative $x$ coordinate because the direction of $\alpha$ provokes a direction with a negative x value. this means, for example, that 0 is the lowest position in the axis $y$ that the gripper can get and 1 is the highest position that the gripper can reach in the axis $y$. These boundaries are given by the robotic arm dimensions, for example, suppose that each link measure 1 unit and we have three links, then the maximum height that is possible to make is 3 units seen from the base of the robotic arm to the gripper and height that is possible to make is 0 units seen from the base.

Therefore, to represent a coordinate which was touched on the LCD touch screen, four conditions are presented which the values of $x_p$ and $y_p$ are greater or equal than to each other, it means, two cases where $x_p \geq y_p$ or $y_p \geq x_p$ occur. The four conditions are represented in four right triangles which represent the workspace of the LCD touch screen. The conditions and its equations are described in [44]. These conditions are the baseline to calculate the inverse-kinematics.

### 4.3. Linear Transformation

Three different values have been defined, which are obtained from the LCD touch screen; then the robotic arm needs to get the point in the space that was touched in the LCD touch screen, so a linear transformation is required to know where the gripper is. It

can be constructed a vector $(x_p, y_p, \alpha)$ which the linear transformation work with it, also $x_p$ is defined such as depth-location of the gripper point, $y_p$ is the height location of the gripper point and $\alpha$ is the direction where $x_p$ and $y_p$ are located such as it was defined in Figure 1a. In summary, when a point in the LCD touch screen is touched, this point is seen from the top view and then, the linear transformation is used to see the same point but with side view. So, the linear transformation is defined first: $T : \mathbb{R}^3 \to \mathbb{R}^3$ and $k$ whatever real number to avoid collision between links for when $\alpha$ rotates counterclockwise such as is presented in Figure 1a and $\alpha$ is defined in [44]:

$$T \begin{pmatrix} x_p \\ y_p \\ \alpha \end{pmatrix} = \begin{pmatrix} kx_p \\ ky_p \\ \alpha \end{pmatrix} \tag{2}$$

If $\alpha$ has to rotate clockwise, so the second linear transformation is used.

$$T \begin{pmatrix} x_p \\ y_p \\ \alpha_p \end{pmatrix} = \begin{pmatrix} -kx_p \\ ky_p \\ -\alpha_p + 180 \end{pmatrix} \tag{3}$$

That means $\alpha$ needs to rotate inversely; therefore, the conditions are switched of its place, the first condition to the fourth condition and the second condition to the third condition, such as it is shown in Figure 1b and $\alpha_p$ is the direction of $(x_p, y_p)$ where the coordinate is located in the condition to arrive.

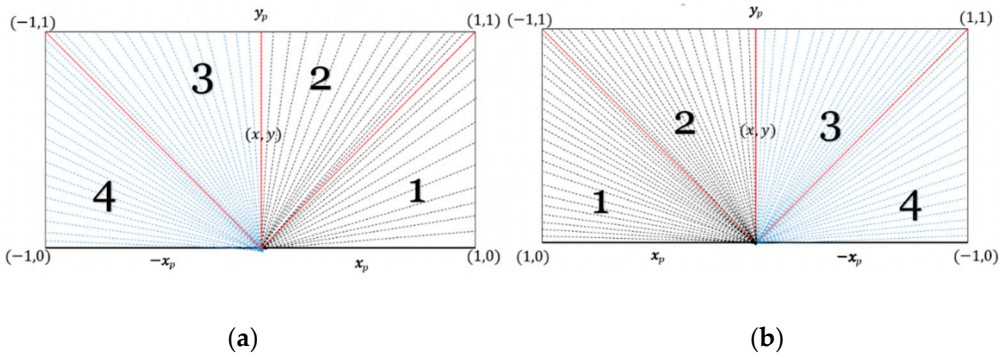

**Figure 1.** Representation of the four conditions and the two linear transformation. (**a**) First linear transformation. (**b**) Second linear transformation.

For example, an initial coordinate $(-x_0, y_0)$ is given which is located in the fourth condition from Figure 1a and $135 \leq \alpha_0 = \tan^{-1}\left(\frac{y_0}{-x_0}\right) + 180 \leq 180$ then $-45 \leq \tan^{-1}\left(\frac{y_0}{-x_0}\right) \leq 0$ and $\alpha'_0 = \tan^{-1}\left(\frac{y_0}{x_0}\right)$ is selected, then $45 \geq -\alpha'_0 = -\tan^{-1}\left(-\frac{y_0}{x_0}\right) \geq 0$ which is obtained that $45 \geq \alpha'_0 = \tan^{-1}\left(\frac{y_0}{x_0}\right) \geq 0$ that means $x_0 \geq y_0$ and the third condition is the final position that is wanted to get to, then observing it in the second condition from Figure 1b, then $x_1 \leq y_1$ and $\alpha_1 = \begin{cases} \tan^{-1}\left(\frac{y_1}{x_1}\right) \\ 90 \end{cases}$, note that $\alpha_0 \neq \alpha_1$ and $\alpha'_0 \neq \alpha_1$, then, applying (3).

$$T \begin{pmatrix} -x_1 \\ y_1 \\ \alpha_1 \end{pmatrix} = \begin{pmatrix} -k(-x_1) \\ ky_1 \\ -\alpha_1 + 180 \end{pmatrix} = \begin{pmatrix} k(x_1) \\ ky_1 \\ -\alpha_1 + 180 \end{pmatrix} \tag{4}$$

Therefore $(kx_1, ky_1, -\alpha_1 + 180)$ where $\alpha'_1 = -\alpha_1 + 180$ and $x_1 \leq y_1$ was got, then $-\alpha_1 = \begin{cases} -\tan^{-1}\left(\frac{y_1}{x_1}\right) \\ -90 \end{cases}$ that is $-\alpha_1 = \begin{cases} \tan^{-1}\left(\frac{y_1}{-x_1}\right) \\ -90 \end{cases}$ with $|x_1| \leq |y_1|$ and since $\alpha_0 \neq \alpha_1$

and $\alpha'_0 \neq \alpha_1$ that means $45 \leq \alpha_1 = \begin{cases} \tan^{-1}\left(\frac{y_1}{x_1}\right) \\ 90 \end{cases} \leq 90$ seen from Figure 1b, multiplying

by $-1$ then $-45 \geq -\alpha_1 = \begin{cases} \tan^{-1}\left(\frac{y_1}{-x_1}\right) \\ -90 \end{cases} \geq -90$ and adding 180 it is had $135 \geq -\alpha_1 +$

$180 = \begin{cases} \tan^{-1}\left(\frac{y_1}{x_1}\right) \\ 90 \end{cases} + 180 \geq 90$, that means $135 \geq \alpha'_1 \geq 90$. Therefore effectively

$(kx_1, ky_1, \alpha'_1)$ is located in the third condition from Figure 1a or in the second condition from Figure 1b and $\alpha'_1$ rotates $\alpha_0 - (-\alpha_1 + 180)$ clockwise to get to third condition from Figure 1a.

Another example, an initial coordinate $(-x_0, y_0)$ is given which is in the third condition seen from Figure 1a and the second condition is the final position that is wanted

to get to, then it is had $y_0 \geq -x_0$ and $\alpha_0 = \begin{cases} \tan^{-1}\left(\frac{y_0}{-x_0}\right) \\ -90 \end{cases} + 180$ then $90 \leq \alpha_0 \leq 135$

where $45 \leq \alpha'_0 = \tan^{-1}\left(\frac{y_0}{x_0}\right) \leq 90$ is obtained and $y_0 \geq x_0$. Calculating the new point $(x_1, y_1)$ which is in the second condition from Figure 1a and calculating with the linear transformation (3) it is had a $\alpha_1 = \tan^{-1}\left(\frac{y_1}{x_1}\right)$ and $y_1 \geq -x_1$ is needed due to $y_1$ and $x_1$ are located in the third condition seen from Figure 1b.

Then:

$$T\begin{pmatrix} x_1 \\ y_1 \\ \alpha_1 \end{pmatrix} = \begin{pmatrix} -k(x_1) \\ ky_1 \\ -\alpha_1 + 180 \end{pmatrix} = \begin{pmatrix} k(-x_1) \\ ky_1 \\ -\alpha_1 + 180 \end{pmatrix} \qquad (5)$$

where $(-kx_1, ky_1, -\alpha_1 + 180)$ is obtained and $\alpha'_1 = -\alpha_1 + 180$ is defined with $x_1 \leq y_1$, that means $\alpha_0 = \alpha'_1$, then $90 \leq \alpha'_1 \leq 135$ multiplying by -1 and adding 180, it is got $90 \geq -\alpha_1 + 180 \geq 45$, then $90 \geq \alpha_1 = \alpha'_0 \geq 45$ with $y_1 \geq x_1$, therefore $(-kx_1, ky_1, \alpha'_1)$ is located in the third condition seen from Figure 1b or $(kx_1, ky_1, \alpha_1)$ second condition that is seen from Figure 1a, then, $\alpha$ (first-joint degree) has to rotate $\alpha_0 - (-\alpha_1 + 180)$ clockwise starting to $\alpha_0$ and getting $-\alpha_1 + 180$ which is in the second condition from Figure 1a. In a similar way, the other remaining cases can be proven and for when the first joint rotates counterclockwise, the (2) linear transformation is used and $0 \leq \alpha = \alpha_p \leq 45$ in the first condition, $45 \leq \alpha = \alpha_p \leq 90$ in the second condition, $90 \leq \alpha = \alpha_p + 180 \leq 135$ in the third condition, and $135 \leq \alpha = \alpha_p + 180 \leq 180$ in the fourth condition and $\alpha$ is defined in [44] for each condition.

In this way, the degree's condition can be found with the second linear transformation, and it can find the degrees that the first-joint has been moved with the coordinate that has been touched. Having said that, it is used the geometry method from [45] to calculate the inverse-kinematics. The equations were represented in the most convenient way to obtain results. Then, the equations are presented below:

$$q_2 = \begin{cases} \tan^{-1}\left(\frac{y_p}{x_p}\right) - \tan^{-1}\left(\frac{l_2 \sin q_3}{l_1 + l_2 \cos q_3}\right) \\ \tan^{-1}\left(\frac{y_p}{x_p}\right) + \tan^{-1}\left(\frac{l_2 \sin q_3}{l_1 + l_2 \cos q_3}\right) \end{cases} \qquad (6)$$

And:

$$q_3 = \begin{cases} \cos^{-1}\left(\frac{x_p^2 + y_p^2 - l_2^2 - l_3^2}{2l_2 l_3}\right) \\ -\cos^{-1}\left(\frac{x_p^2 + y_p^2 - l_2^2 - l_3^2}{2l_2 l_3}\right) \end{cases} \qquad (7)$$

where $q_2$ and $q_3$ are the final position of the second-joint and third-joint and $l_2 = 1$ and $l_3 = 1$ are the length of the second link and third link. This inverse kinematic is for a 2 DOF robotic arm, we can use these formulas to get the position $(x_p, y_p)$ with a direction of $\alpha$ which represents the degrees that the first-joint has to rotate. This algorithm can be implemented to a robotic arm with more degrees of freedom as long as exist a geometric

method to calculate the degree of the second-joint to the n-joint. For example, supposing that a 6DOF manipulator is used, the formulas could be used to calculate the inverse kinematic of 5DOF manipulator presented in [46].

### 4.4. Distribution and Trajectory Function

The next distribution function has been proposed because is too easy to calculate the trajectory for whatever point; moreover, it is exact, due to the degree or radians are ordained in the whole interval $[0,1]$ which is the number $S_f$ the final position, so, if the final position is 180 degree, that means $S_f = 1$, or if the final position is 45 that means $S_f = 1/4$. In this way, the radians or degrees between 0 and $\pi$ radians (180 degrees) are represented in the interval $[0,1]$ with the number $S_f$ with whatever time of interest $(t_s)$. The distribution function with parameters $\zeta, a$ and $t_s$ is represented below.

$$S(x) = \begin{cases} \dfrac{n^{x-\zeta}}{\left| n^{x+\frac{t_s}{x}-1} - n^a \right|} & \text{where } a < t_s \text{ and } 0 < x \\ 0 & \text{if } x \le 0 \end{cases} \tag{8}$$

where $n$ is a natural number greater than 1, $n > 1$, it can help to reduce the value of the parameter $\zeta$, and different velocities, accelerations, and different shapes of the trajectory can be obtained.

Moreover, the function $S(x)$ is a distribution function because it fulfils the characteristics of a distribution function that has been proved in [47]. Once the distribution function-($\zeta at_s$) has been presented, it is proposed how the trajectory function-($\zeta at_s$) using the Equation (8) must work in order to compute a safe and fast point-to-point trajectory.

The distribution function has to get to the final position $S_f$ in the start position which is $q_0$ where it belongs to $[0,1]$. As it was mentioned before, reducing the compile time is wanted; the computation of the function is necessary to reduce. Also, it is wanted to have an exact trajectory, which means, with high accuracy in the results. Thereby, one parameter is assigned to each join space trajectory; for these reasons, the next characteristic is introduced. Exist a number $\zeta$ for whatever $x > 0, q_0$ and $S_f$ such that $S_f \ge S(x) + q_0$ and $S(x) + q_0 > q_0$, and for when $x$ gets to $t_s$, then the equalization is fulfilled:

$$S(x) = \frac{n^{x-\zeta}}{n^{x+\frac{t_s}{x}-1} - n^a} > 0 \tag{9}$$

then:

$$q_0 < \frac{n^{x-\zeta}}{n^{x+\frac{t_s}{x}-1} - n^a} + q_0 \le S_f \tag{10}$$

Then $\zeta \ge x - \dfrac{\ln\left[ (S_f - q_0)\left( n^{x+\frac{t_s}{x}-1} - n^a \right) \right]}{\ln n}$, and for when $x$ is approaching to $t_s$ $(x \to t_s)$, occurs:

$$\zeta = \lim_{x \to t_s} x - \frac{\ln\left[ \left( S_f - q_0 \right)\left( n^{x+\frac{t_s}{x}-1} - n^a \right) \right]}{\ln n} = t_s - \log_n\left[ \left( S_f - q_0 \right)\left( n^{t_s} - n^a \right) \right] \tag{11}$$

For when $S_f \le -\dfrac{n^{x-\zeta}}{\left| n^{x+\frac{t_s}{x}-1} - n^a \right|} + q_0 < q_0$, then:

$$\zeta = t_s - \frac{\ln\left[ \left( q_0 - S_f \right)\left( n^{t_s} - n^a \right) \right]}{\ln(n)} = t_s - \log_n\left[ \left( q_0 - S_f \right)\left( n^{t_s} - n^a \right) \right] \tag{12}$$

In this way, the trajectory is made from an initial position greater than the final position, which means $q_0 > S_f$.

Using the Equation (8) and adding $q_0$, the trajectory function-($\zeta at_s$) is defined and used for calculating the linear and curved motion trajectory in order to obtain the final position from the LCD touch screen. The trajectory function-($\zeta at_s$) is presented below:

$$S_{\zeta at_s}(x) = \begin{cases} q_0 \; if \; x \leq 0 \\ \dfrac{n^{x-\zeta}}{\left| n^{x+\frac{t_s}{x}-1}-n^a \right|} + q_0 \; if \; 0 < x \leq t_s \; and \; q_0 \leq S_f \quad (1st \; condition) \\ -\dfrac{n^{x-\zeta}}{\left| n^{x+\frac{t_s}{x}-1}-n^a \right|} + q_0 \; if \; 0 < x \leq t_S \; and \; S_f < q_0 \quad (2nd \; condition) \end{cases}$$

(13)

where $n$ is a real number such that $n > 1$. When $n$ is so close to 1, large values of $\zeta$ are presented and that could provoke great magnitudes of velocity, acceleration and Jerk. Then, $n = 2$ is used to take advantages of non-large values of $\zeta$ and allows us to show other advantages for when $t_s = 1$. Then $n = 2$ is used to get the results using the algorithm proposed and show some examples to obtain different shape of trajectories, implementing the function (13) and calculating the parameter $\zeta$ with (11) and (12).

Example 1: It is used Equation (11) to compute $\zeta$ using $n = 2$ and $n = 12$ with $t_s = 4$ and $q_0 = 0.45$, two different trajectories are generated with the same end-point $S_f = 0.8$ calculated by (13: 1st condition) such as it is shown in Figure 2.

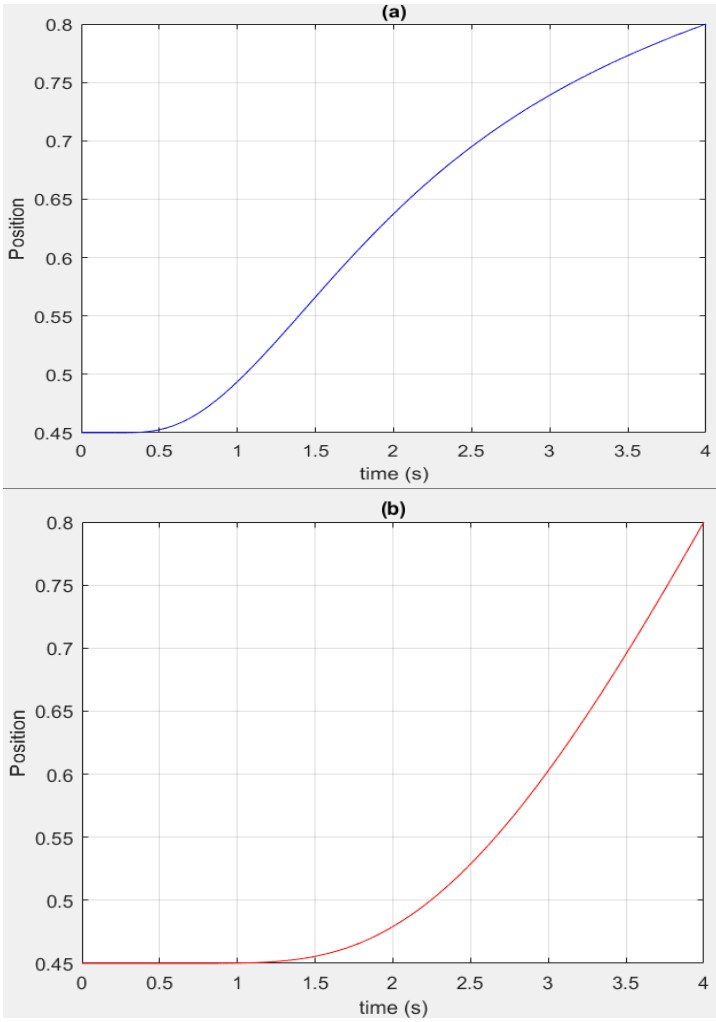

**Figure 2.** (**a**) presents the trajectory $S_{\zeta at_s}(x)$ for when $n = 2$ and $n = 12$ (**b**) in order to obtain a different shape of the trajectory planning.

Example 2: When the 2nd condition is used, then, $\zeta$ is calculated by using the Equation (12). Then, a decreasing trajectory is presented. In this example the final position is $-0.85$ and the trajectory starts in 0.65, this means $S_f < q_0$. Figure 3 represents the trajectory with a total time equal to 1 and using a parameter $a = 0$ and $\zeta = 0.53915881$.

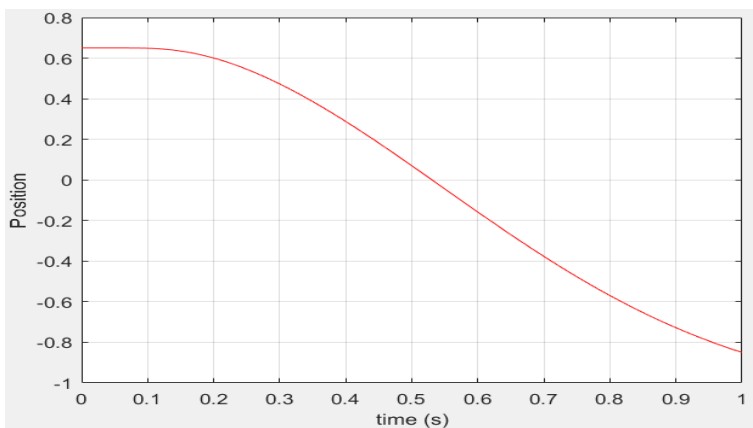

**Figure 3.** Representation of the second case.

Example 3: Different shapes of trajectory have been presented at this moment, but this characteristic can increase if the parameter $a$ is changed (see Figure 4). Moreover, the velocity, acceleration, and Jerk can increase and decrease in order to obtain linear or curved trajectories as well as diversity to generate trajectories.

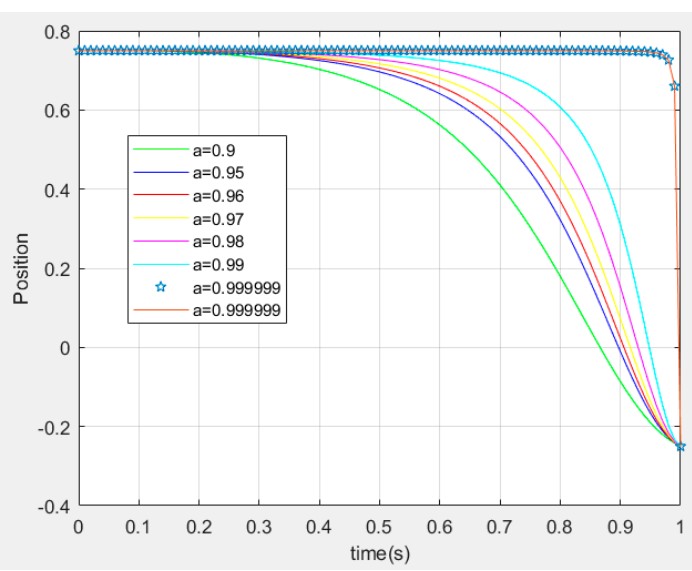

**Figure 4.** Different examples of trajectories for when $a$ approximates to $t_s = 1$.

This property is only presented when the total time of the trajectory is equal to 1 because $\left| n^{(x+\frac{1}{x}-1)} - n^a \right|$ is not enough small for when $x$ is $0 < x < t_s$ (the reader can draw the function $g(x) = n^{x-\zeta}$ and $h(x) = n^{(x+\frac{1}{x}-1)} - n^a$ to get a better idea). However, a curved motion can be obtained even when the total time is greater than 1, for example in Figure 5, different curved motions are shown.

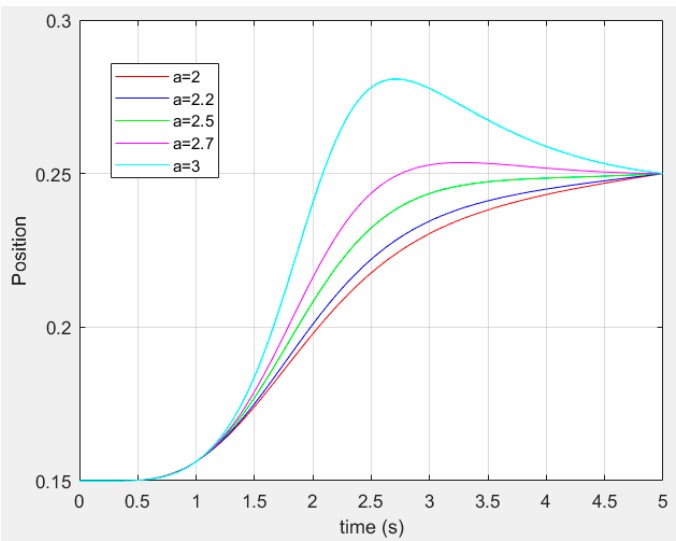

**Figure 5.** Different examples of trajectories for different values of *a* and a total time equal to 5.

### 4.5. Calculation of Velocity and Acceleration

Finally, in order to generate a safe and fast trajectory, the velocity and acceleration calculation must be obtained. Getting the position from $S(x)$ it can be obtained $\dot{S}(x) = \frac{d}{dx}S(x)$ and $\ddot{S}(x) = \frac{d^2}{dx^2}S(x)$ to obtain the velocity and acceleration respectively.

The velocity is represented below:

$$\dot{S}(x) = \begin{cases} \dfrac{\ln|n|n^{x-\zeta+1}\left(t_s n^{x+\frac{t_s}{x}} - n^{a+1}x^2\right)}{\left(n^{x+\frac{t_s}{x}} - n^{a+1}\right)^2 x^2} \\ 0 \; if \; x \leq 0 \end{cases} \tag{14}$$

Note that $\dot{S}(x)$ is effectively a density function, according to [40]; therefore, $\dot{S}(x)$ is always positive $\dot{S}(x) \geq 0$, only in cases that the distribution function is used as a trajectory function and $S(x)$ takes negative values (for example $q_0 < 0$ or $S_f < 0$), the density function that is to say the velocity function can be positive and pass to a negative values.

Then, according to [48], the acceleration is represented below:

$$\ddot{S}(x) = \begin{cases} \dfrac{\ln(n)n^{x-\zeta+1}\left(\left(n^{2x+\frac{2t_s}{x}}\ln(n)+n^{x+\frac{t_s}{x}+a+1}\ln(n)\right)t_s^2 + \left(n^{x+\frac{t_s}{x}+a+1}\ln(n)+n^{2a+2}\ln(n)\right)x^4 + \left(2n^{x+\frac{t_s}{x}+a+1}-2n^{2x+\frac{2t_s}{x}}\right)t_s x - 4n^{x+\frac{t_s}{x}+a+1}\ln(n)t_s x^2\right)}{\left(n^{x+\frac{t_s}{x}} - n^{a+1}\right)^3 x^4} \\ 0 \; if \; x \leq 0 \end{cases} \tag{15}$$

Then, the position, velocity, acceleration and Jerk (the derivative of the acceleration) is obtained and shown in Figure 6 for when $t_s = 1$ with different value of *a*.

Changing the value of *a*, a smooth trajectory can be obtained. Note that the velocity, acceleration and Jerk start at 0 and also these functions are finites, moreover, the Jerk is continuous and can approximate to 0 for using different values of *a*. This is an advantage characteristic when the function $S_{\zeta a t_s}(x)$ is used because it generates smooth trajectories, and we can take care of the robotic arm quality.

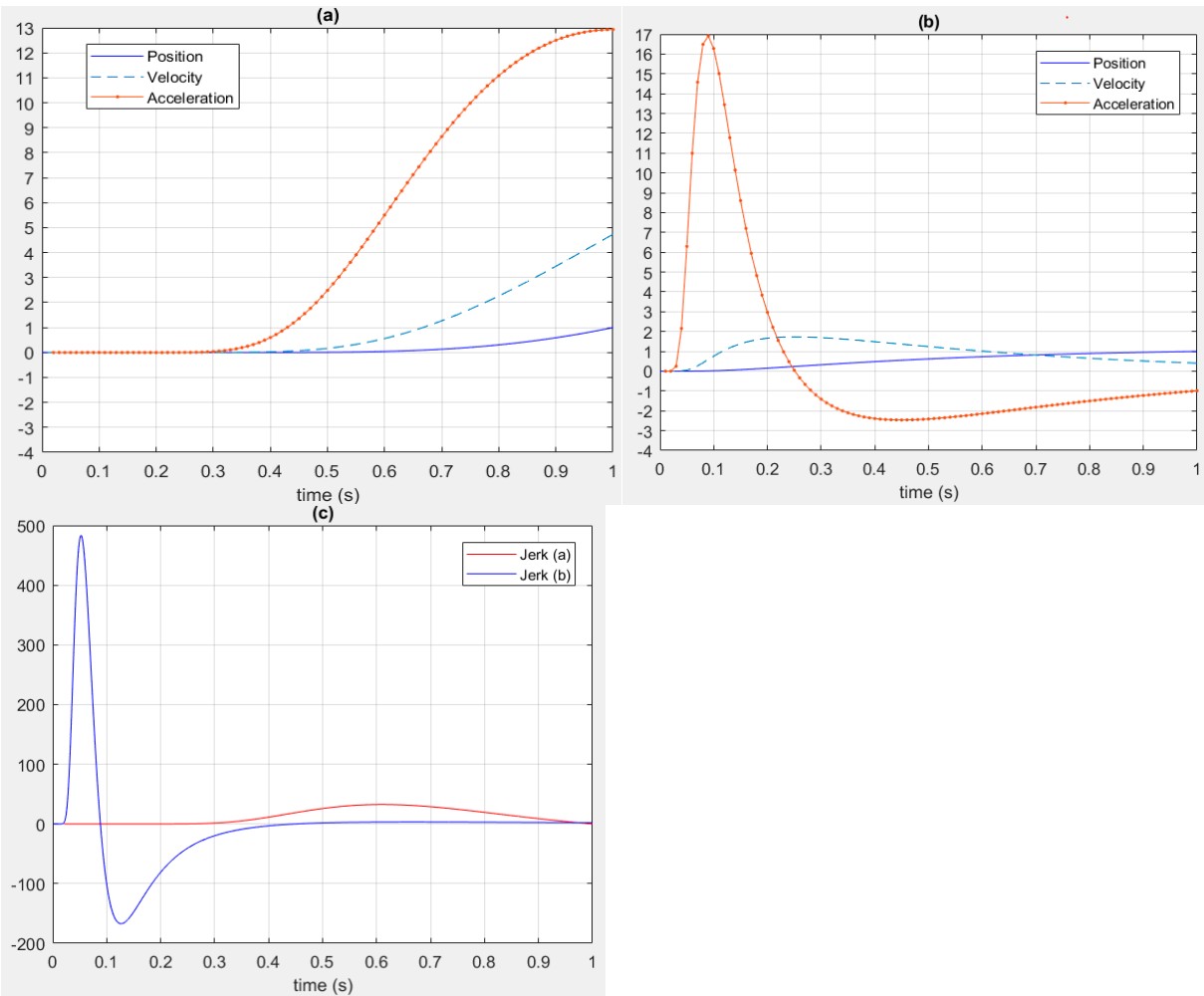

**Figure 6.** (**a**) and (**b**) show position, velocity, and acceleration and (**c**) shows the Jerk of the trajectory in a second, starting in the position 0 and finalizing in 1 using different values of *a* to change the velocity and acceleration (the parameter *a* of (**a**) is greater than the parameter *a* of (**b**)).

To obtain the trajectories in Matlab, the function (13) is multiplied by $\pi$, therefore the units are $\pi - rad/s$ and $\pi - rad/s^2$ for the velocity and acceleration respectively. The position is obtained in radians using the Equations (6) and (7) to obtain the inverse kinematic and the gripper position.

It is necessary to emphasize that the final result using the function (13) and a parameter $\zeta$ with three decimals are not affected in precision (see Figure 7). Also, to generate the desired point-to-point trajectory just require one computation with high exactitude. To show this, a short trajectory is generated using $\zeta = 17.609$ and the resulting trajectory is compared with the resulting trajectory with $\zeta$ calculated by using the function (13), which value is 17.6096404.

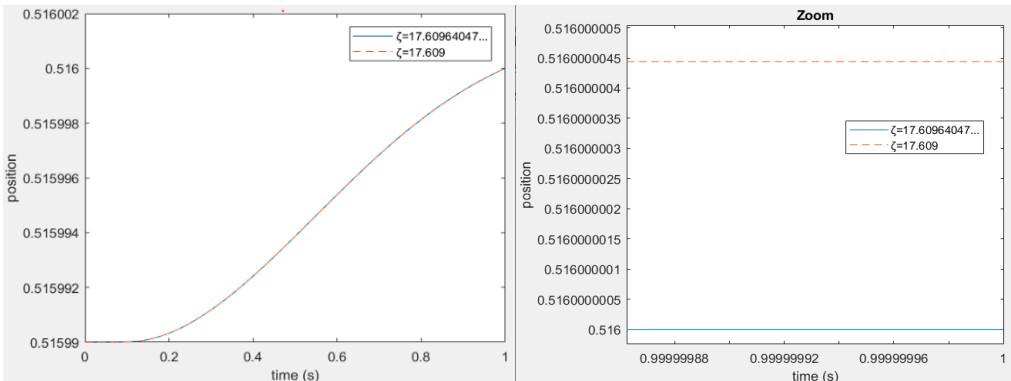

**Figure 7.** Two different results of trajectory to compare the final value of the trajectory, using three decimals and many decimals. On the right side of the figure, an approach is presented to observe the difference between the two final results obtained.

## 5. Simulation and Results

First, the LCD touch screen is used in Arduino Mega and it is connected to MATLAB, where the data is read, then, the simulation and the results obtained are found in the guide of MATLAB. The guide shows the results of position, velocity, and acceleration which are in the next tables, with velocity and acceleration graphs. With the coordinates $(x_p, y_p)$, the degrees of the second and third articulation are calculated with the function (9) and (10), and the degrees of the first articulation is calculated depending on the four conditions where the coordinate $(x_p, y_p)$ are located. Secondly, the degrees are switched $(q_2, q_3, \alpha)$ in the closed interval $(0, 1)$ or $(-1, 0)$, and finally, the trajectory is calculated, with the current position encrypted and calculating the parameter $\zeta$. In Figure 8, the whole process and position-zero are represented.

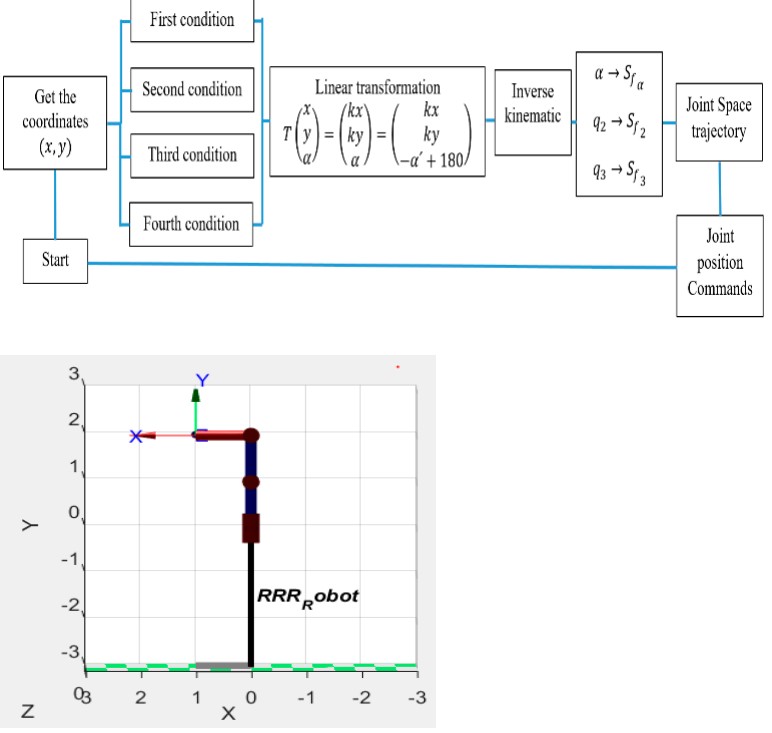

**Figure 8.** Illustration of the whole process to calculate the trajectory and the position-zero of the robotic arm where $q_1, q_2, q_3 = 0$ wich is presented on the right side and in the task space, the position is $x_p = 1$, $y_p = 2$ and $\alpha = 0$, so it is had the coordinate (2, 0) with a direction equal to 0.

This process is repeated the times as much as the trajectory is calculated with the LCD touch screen and the initial position is $q_0 = 0$ in each joint (that means $q_1, q_2$ and $q_3$ are equal to zero, which are presented in Figure 8). In Table 1, $\alpha$ is between $0 \le \alpha \le 45$ degrees and $0 \le S_{f\alpha} \le 0.25$, with the different coordinates $(x_p, y_p)$ represented as a division of $y_p$ over $x_p$, and the value of each $\zeta$, velocity and acceleration of the joint-2, joint-3 and joint-$\alpha$ (or joint-1). In Figure 9 the position, velocity, and acceleration vectors are shown for the first condition, the trajectory animation is shown using the example of the last result from the Table 1 with $a_2 = -1$, $a_3 = -5$ and $a_\alpha = -2.5$ and the gripper is in the end-point (1,1,45) according to the linear transformation in the Equation (2).

**Table 1.** Results of the trajectory planning in the first condition, the coordinate $(x_p, y_p)$ is located in the first right triangle's area.

| $\alpha$. | $\left(\frac{y_p}{x_p}\right)$ | $(\zeta_2, \zeta_3, \zeta_\alpha)$ | $T(S_2(x), S_3(x), S_\alpha(x))$ | $\dot{S}(x)$ | $\ddot{S}(x)$ |
|---|---|---|---|---|---|
| 5.71 | $\frac{0.075}{0.75}$ | (2.534, 1.407, 5.978) | (−0.345, 0.75, 0.031) | (0.527, 1.152, 0.048) | (2.0371, 4.682, 0.187) |
| 11.3 | $\frac{0.17}{0.85}$ | (5.299, 5.666, 6.009) | (−0.294, 0.714, 0.062) | (0.077, 0.060, 0.047) | (0.3011, 0.2126, 0.177) |
| 21.8 | $\frac{0.415}{0.83}$ | (4.386, 6.521, 5.099) | (−0.198, 0.692, 0.121) | (0.146, 0.0333, 0.089) | (0.5669, 0.1299, 0.35) |
| 30.96 | $\frac{0.47}{0.78}$ | (6.510, 8.332, 5.296) | (−0.176, 0.699, 0.172) | (0.033, 0.009, 0.077) | (0.129, 0.041, 0.3005) |
| 41.98 | $\frac{0.83}{0.93}$ | (4.023, 3.972, 5.029) | (−0.053, 0.571, 0.233) | (0.188, 0.194, 0.093) | (0.7263, 0.7558, 0.361) |
| 45 | $\frac{1}{1}$ | (5.214, 4.803, 6.897) | (−0.000, 0.500, 0.250) | (0.082, 0.109, 0.025) | (0.313, 0.4192, 0.0991) |

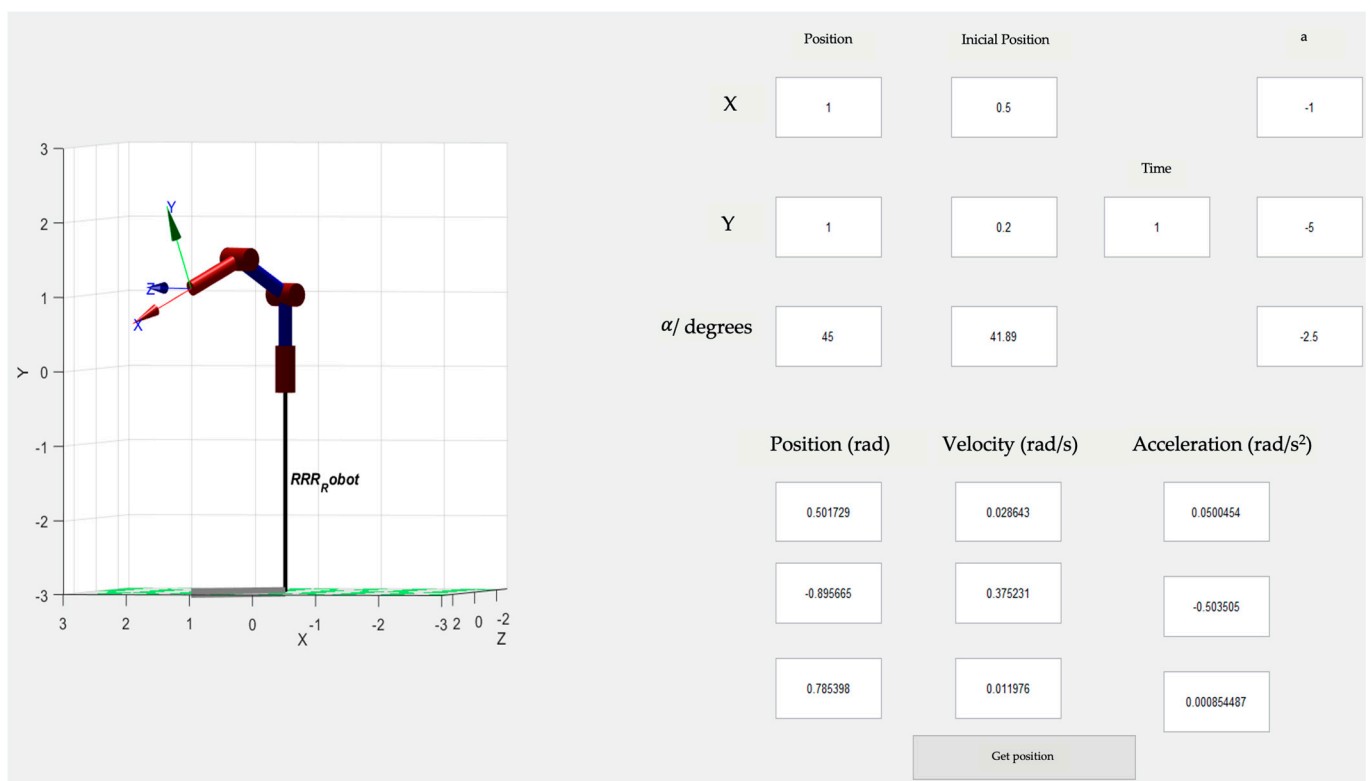

**Figure 9.** Guide of MATLAB, where the final position (rad), velocity (rad/s) and acceleration (rad/s$^2$) values are shown, and the trajectory is seen with the joint position commands.

In Figure 10 the velocity functions have the maximum value in the same point of the domain of $\dot{S}(x)$, which means, a linear-motion is obtained. In Table 2 the results of the second condition are shown, taking deferent values greater than zero of $a_2, a_3$ and $a_\alpha$ to generate a curved-motion and starting in $q_0$ for when $\alpha = 45$.

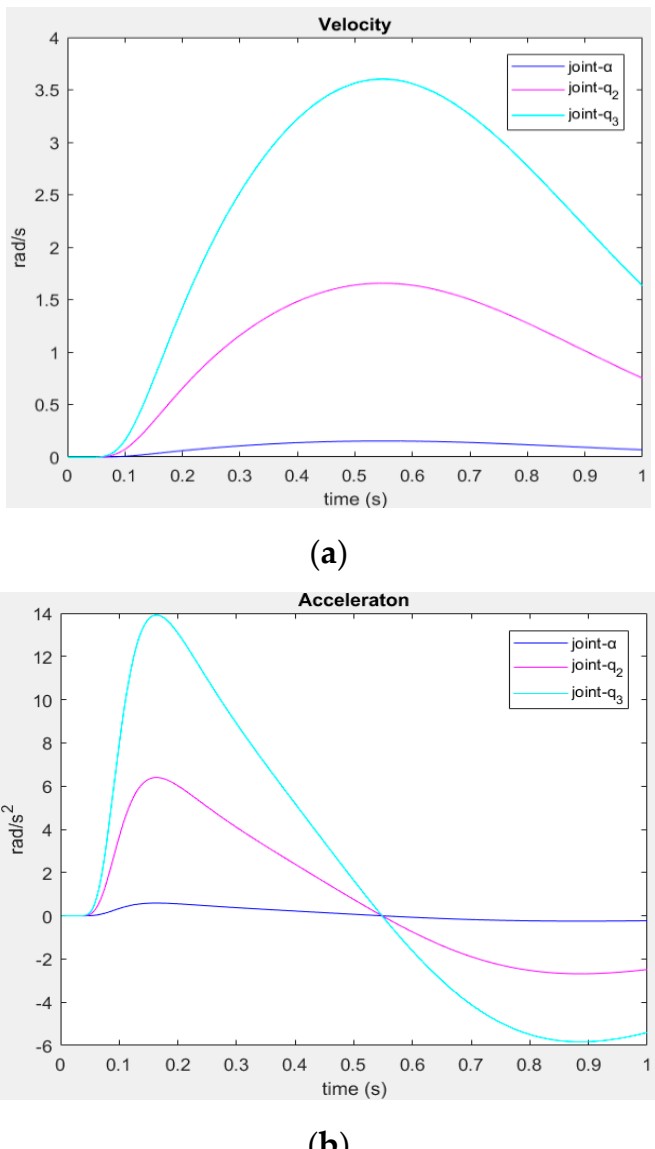

**Figure 10.** Example of a velocity and acceleration acquired of a linear-motion on each joint in the first condition with $a = 0$. The velocity and acceleration are multiplied by $\pi$ to get the units (rad/s and rad/s$^2$ respectively). (**a**) Velocity. (**b**) Acceleration.

**Table 2.** Results of the trajectory planning in the second condition, the coordinate $(x_p, y_p)$ is located in the second right triangle's area and the three joints start with $q_1 = q_2 = q_3 = 0$ in the first result for when $\alpha = 45$ degrees.

| $\alpha$ | $\left(\frac{y_p}{x_p}\right)$ | $(\zeta_2, \zeta_3, \zeta_\alpha)$ | $T(S_2(x), S_3(x), S_\alpha(x))$ | $\dot{S}(x)$ | $\ddot{S}(x)$ |
|---|---|---|---|---|---|
| 45 | $\left(\frac{0.82}{0.82}\right)$ | (2.162, 2.0277, 3.312) | (−0.053, 0.606, 0.250) | (0.0825, 0.1651, 0.399) | (0.5553, 1.110, 2.6193) |
| 63.434 | $\left(\frac{0.2}{0.4}\right)$ | (2.162, 2.0227, 3.312) | (−0.280, 0.856, 0.352) | (0.3535, 0.389, 0.159) | (2.378, 2.619, 1.071) |
| 71.56 | $\left(\frac{0.93}{0.31}\right)$ | (2.184, 2.4501, 4.463) | (0.060, 0.673, 0.397) | (0.579, 0.299, 0.0708) | (2.3352, 1.942, 0.4812) |
| 80.53 | $\left(\frac{0.77}{0.128}\right)$ | (6.058, 3.816, 4.3267) | (0.075, 0.744, 0.447) | (0.023, 0.1109, 0.079) | (0.1596, 0.7536, 0.529) |
| 90 | $(x_p \approx 0, y_p = 1)$ | (3.680, 3.458, 4.2485) | (0.166, 0.666, 0.500) | (0.142, 0.1218, 0.082) | (0.9659, 0.8279, 0.558) |

The maximums of each acceleration are located almost in the same location at the domain of $\dot{S}(x)$, then, and a curve-motion is presented due to the maximum values of the acceleration are not close and the acceleration functions are different distributed through the time (see Figure 11). Table 3 shows the results of the trajectory in the third condition, using $-100 \le a_2 = a_3 = a_\alpha < 0$.

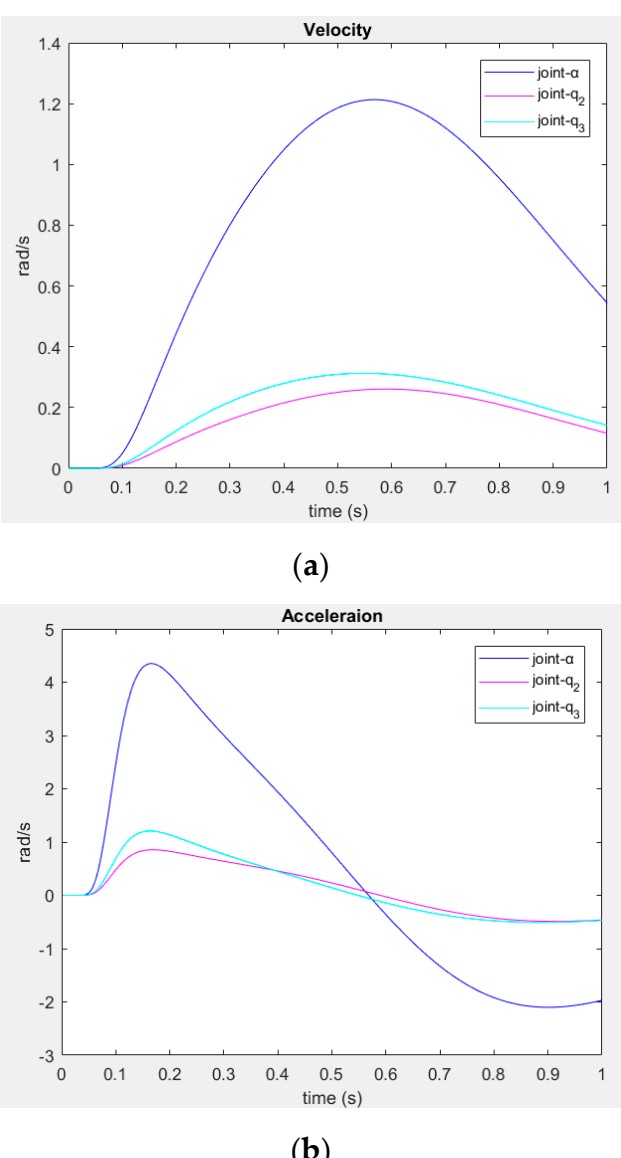

**Figure 11.** Example of a velocity and acceleration acquired of a curved-motion on each joint in the second condition using $a_2, a_3, a_\alpha > 0$. (**a**) Velocity. (**b**) Acceleration.

**Table 3.** The trajectory planning results in the third condition are shown, the coordinate $(x_p, y_p)$ is located in the third right triangle's area and the three joints start with $q_1 = q_2 = q_3 = 0$ in the first result for when $\alpha = 90$.

| $-\alpha + 180$ | $\left(\frac{y_p}{x_p}\right)$ | $(\zeta_2, \zeta_3, \zeta_\alpha)$ | $T(S_2(x), S_3(x), S_\alpha(x))$ | $\dot{S}(x)$ | $\ddot{S}(x)$ |
|---|---|---|---|---|---|
| 90 | $(x_p \approx 0, y_p = 0.61)$ | (3.271, 0.335, 1.000) | (0.103, 0.792, 0.500) | (0.161, 1.238, 0.781) | (0.568, 1.136, 4.511) |
| 100.8 | $(\frac{0.67}{-0.127})$ | (4.257, 6.143, 4.058) | (0.051, 0.778, 0.560) | (0.081, 0.022, 0.093) | (0.556, 0.150, 0.636) |
| 108.434 | $(\frac{0.56}{-0.186})$ | (4.111, 5.019, 4.559) | (−0.006, 0.809, 0.602) | (0.090, 0.048, 0.066) | (0.614, 0.327, 0.450) |
| 115.2 | $(\frac{0.69}{-0.32})$ | (7.097, 4.115, 4.733) | (−0.014, 0.751, 0.640) | (0.011, 0.090, 0.058) | (0.077, 0.612, 0.399) |
| 135 | $(\frac{0.97}{-0.97})$ | (7.753, 2.102, 3.184) | (−0.009, 0.518, 0.750) | (0.007, 0.363, 0.171) | (0.049, 2.471, 1.167) |

According to Table 3, most of the time, a linear motion is presented, and also, the trajectory is smooth and equilibrated (see Figure 12).

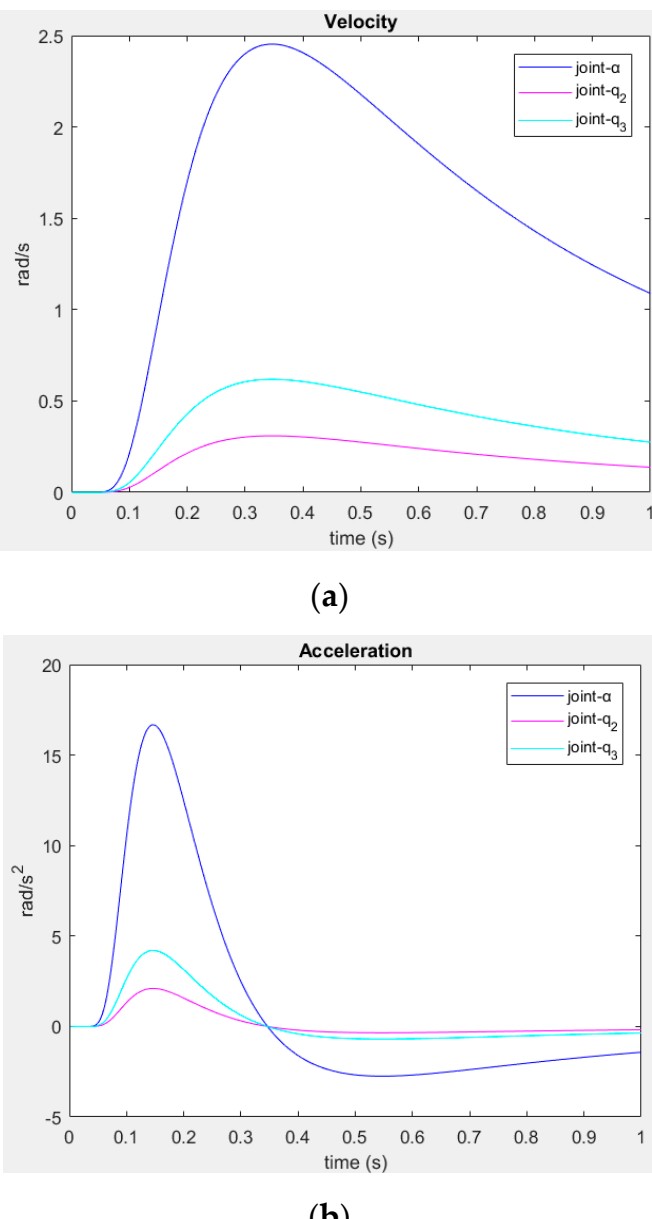

**Figure 12.** Example of a velocity and acceleration acquired on each joint in the third condition, using $a_2 = a_3 = a_\alpha < 0$ to obtain a smoother linear-motion. (**a**) Velocity. (**b**) Acceleration.

In Table 4, the fourth condition is showed; it is had a $-1.5 \leq a_2 = a_3 = a_\alpha \leq 0$ to observe the velocity and acceleration behavior, the numbers $a_2, a_3$, and $a_\alpha$ have been changed through the fourth condition.

**Table 4.** Results of the trajectory planning in the fourth condition, the coordinate $(x_p, y_p)$ is located in the fourth right triangle's area and the three joints start with $q_1 = q_2 = q_3 = 0$ in the first result for when $\alpha = 135$ and in the last result, $\alpha = 180$.

| $-\alpha + 180$ | $\left(\frac{y_p}{x_p}\right)$ | $(\zeta_2, \zeta_3, \zeta_\alpha)$ | $T(S_2(x), S_3(x), S_\alpha(x))$ | $\dot{S}(x)$ | $\ddot{S}(x)$ |
|---|---|---|---|---|---|
| 135 | $\left(\frac{0.81}{-0.81}\right)$ | (4.442, 0.989, 0.695) | (−0.055, 0.611, 0.750) | (0.084, 0.927, 1.136) | (6.6471, 5.802, 7.121) |
| 141.34 | $\left(\frac{0.595}{-0.74}\right)$ | (4.229, 4.184, 5.242) | (−0.127, 0.685, 0.785) | (0.106, 0.110, 0.053) | (1.193, 1.196, 1.172) |
| 153.43 | $\left(\frac{0.41}{-0.82}\right)$ | (4.388, 6.821, 4.311) | (−0.2008, 0.696, 0.852) | (0.110, 0.017, 0.101) | (1.936, 1.173, 1.189) |
| 171 | $\left(\frac{0.101}{-0.64}\right)$ | (3.422, 4.054, 5.273) | (−0.345, 0.790, 0.950) | (0.216, 0.139, 0.060) | (1.939, 1.930, 1.929) |
| 180 | $\left(\frac{0}{-0.19}\right)$ | (4.005, 3.742, 5.321) | (−0.4697, 0.9394, 1.00) | (0.190, 0.228, 0.076) | (1.931, 1.943, 1.949) |

As can be noted in Figures 10–13, different motions are presented because the parameter *a* has been changed through the four conditions. The maximums of the acceleration function are almost located closet of each other in the domain of $\ddot{S}(x)$ such as it was presented in Table 4. Most of the time, a smooth trajectory is obtained (this depends on the values *a* and *n* chosen) because the velocity is suitably distributed through the time and, therefore, the acceleration too (see Figure 13). Therefore, the velocity has to be conveniently distributed through the time to have a smoother trajectory using the parameter *n* and *a*.

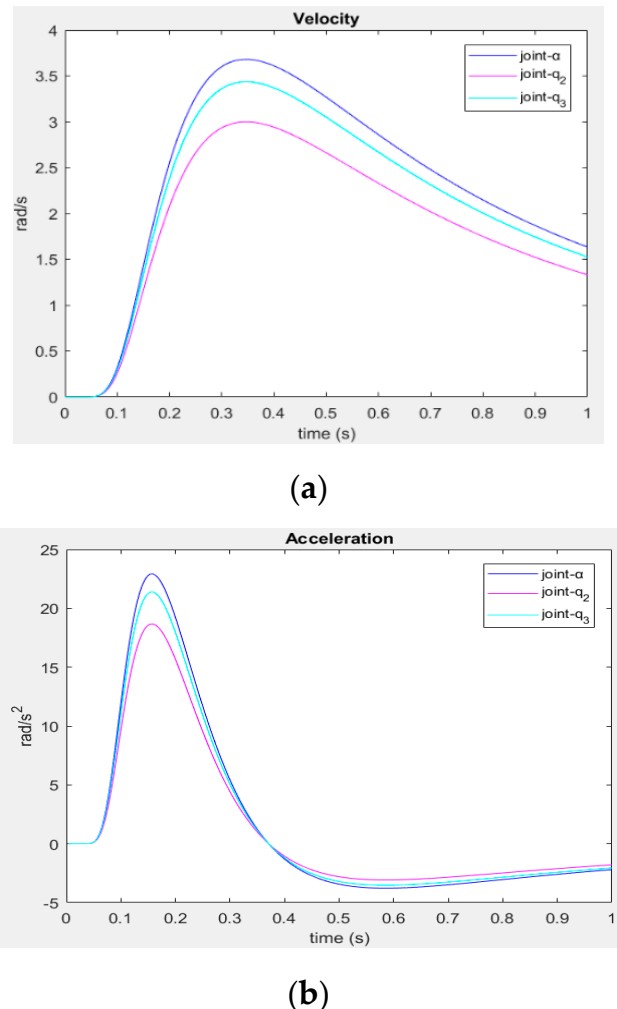

**(a)**

**(b)**

**Figure 13.** Example of a velocity and acceleration acquired on each joint in the fourth condition using $-1.5 \leq a_2 = a_3 = a_\alpha \leq 0$ to obtain a smooth not curved-motion. (**a**) Velocity. (**b**) Acceleration.

## 6. Discussion

The trajectory obtained in the four conditions is exact and comfortable to obtain a safe and fast trajectory. It is not necessary to calculate and make restrictions such as polynomials trajectory, and it is not presented oscillations and inexactitudes in the end-point. Moreover, the general problem of all methods to calculate the trajectory is the presentation of oscillations and using the distribution function presented; the oscillation is eliminated in any case. Also, with this algorithm, it has been decreased the process of calculation and hence the compile time reduced.

The velocity and acceleration calculated using this distribution function can be modified according to the desired kind of movement, so, it can be had many options according to obtain the best safe and fast trajectory, unlike other methods. That is, the trajectory has only one option to generate the velocity and acceleration for a specific start-point and

end-point, and most of the time, oscillations occur in large jerks, and the accuracy of the results decreases in smaller trajectories.

It is important to highlight that the algorithms mentioned in the related works, in addition to using the algorithm to calculate the trajectory, usually use other techniques such as an additional feedback control algorithm or employ a time-based profile generator to calculates key times, and in this way, to obtain the optimal time calculations and reduce unnatural behavior, with this proposed algorithm no further calculations are necessary, this distribution function only performs a calculation and as a result a faster and safer process is obtained to calculate the trajectory as well as showed in the results, the trajectory is always accurate with a determinate shape of the trajectory according to the parameter (a).

In Table 5, it is presented different end-position of each condition and it has been compared the values of the end-position using the distribution function proposed and cubic polynomial with a trajectory of 1 s. It can be observed that the results of the cubic polynomials are inaccurate while the distribution function is accurate. Figure 14 shows these simulated results in MATLAB with an anthropomorphic robotic arm. It can be clearly seen that the trajectory of the joint -$\alpha$ rotates starting in position zero $q_0 = 0$ and gets to the end-position equal to 1, that means, 180 degrees or $\pi$ radians. It is used the distribution function presented with a total time of 1 s, as it can be noted, the result of the end-position is 3.14159...radians, that means 180 degrees, unlike cubic polynomial, the result of the end-position is 3.0533 radians which is 174.941204 degrees. Therefore, the distribution function proposed is more accurate than cubic polynomials and present a smooth trajectory, according to the velocities and acceleration obtained.

**Table 5.** Results obtained with the algorithm proposed in this work in comparison with the Cubic Polynomial Algorithm.

| $\alpha$ | Trajectory Function-($\zeta at_s$) | Cubic Polynomials |
|---|---|---|
| 45 | $0.25(\pi)$ rad $\equiv 45°$ | 0.7633 radians $\equiv 43.73°$ |
| 71.56 | $0.3976(\pi)$ rad $\equiv 71.5624°$ | 1.2139 radians $\equiv 69.55°$ |
| 115.2 | $0.64(\pi)$ rad $\equiv 115.1988°$ | 1.9541 radians $\equiv 111.9656°$ |
| 180 | $1(\pi)$ rad $\equiv 180°$ | 3.053 radians $\equiv 174.941204°$ |

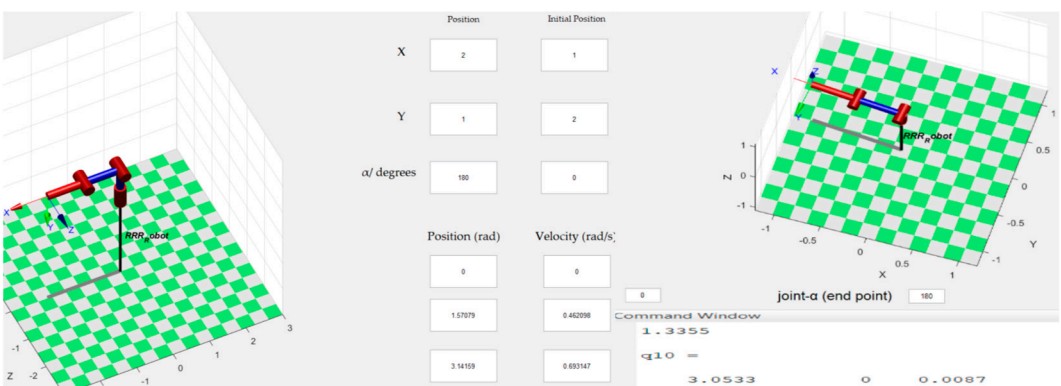

**Figure 14.** Simulation results of the alpha-join trajectory using distribution function proposed and cubic polynomial with a trajectory of 1 s.

Note that the gripper is situated in the coordinate $(-2, 1)$ and the input coordinate is $x = 2$ and y = 1, that means $(x_p, y_p) = (1, 0.25)$ (obtained from the LCD Touch screen) with $\alpha$=180 then $x_p = -1$ seen from condition 4 Figure 1a or $x_p = 1$ (input) seen from condition Figure 1b. Then, $\alpha$ determinates the sense of $x_p$ touched in the LCD Touch screen, no matter which is the sense of the input value of X.

## 7. Conclusions

A novel algorithm for the trajectory calculation of a robotic arm using an LCD touch screen with the distribution function has been proposed. This proposal is easy to carry out to get a safe and fast trajectory results. Moreover, the distribution function is exact when the calculation of $\zeta$ is correct, and just one computation has to be it to obtain the trajectory (see Table 5). The way to calculate $\zeta$ has a simple formula to get the trajectory in an easy way to interpret. The function starts in $q_0$ and gets to $S_f$ when the time is complete $t_s$, similar to the n-order polynomial, but, in a different way to the trajectory method mentioned and whatever another technique. The distribution function calculates the trajectory with one computation which implies a faster process to calculate the trajectory in whatever time of interest. Also, the velocity and the acceleration functions do not implement always the same values, because these can be modified to obtain a smoother trajectory just changing the parameter $a$ or $n$ and it can be obtained a curved or linear motion. The polynomial trajectory increments its coefficients to obtain a smoother trajectory, and the function proposed has the advantage to get the flattest trajectory changing the parameter $a$ and different motions such as linear or curve motion.

In future works, the value of $\zeta$ can be optimized because it can help to make different economics controls and electronics designs such as the control touch presented in this paper, but now, in a more direct way such as the PID controller. Another future work can be developed for adding more joints to the robotic arm using this method and getting an inverse-kinematics in function to the axis X and Y. Also, another interesting future work is using the method proposed in the work of Mercorelli [14] to estimate the velocity and acceleration in the presence of noise and combine the properties and advantages of using the trajectory function-($\zeta a t_s$) presented in this work. Finally, the parameter $\zeta$ could be defined as a variable to obtain many via points with great accuracy and convenient velocity/acceleration functions, but this is another proposed for future work.

**Author Contributions:** Conceptualization, Y.Q. and O.Z. Ideas; formulation or evaluation of general research objectives and goals; methodology, Y.Q., J.M. and C.L. were developed and designed the methodology; as well as creating models; software, Y.Q. and O.Z. Programming, software development; designing computer programs; implementation of the computer code and supporting algorithms; testing of existing code components; validation, Y.Q., J.M. and C.L. Verification, whether as a part of the activity or separate, of the overall replication/reproducibility of results/experiments and other research outputs; formal analysis, Y.Q. and O.Z. Application of statistical, mathematical, computational, or other formal techniques to analyze or synthesize study data; investigation, Y.Q., O.Z. and J.M. Conducting a research and investigation process, specifically performing the experiments, or data/evidence collection; resources, Y.Q., O.Z., J.P. and R.E. Provision of study materials, reagents, materials, patients, laboratory samples, animals, instrumentation, computing resources, or other analysis tools; data curation, Y.Q. and O.Z. Management activities to annotate (produce metadata), scrub data and maintain research data (including software code, where it is necessary for interpreting the data itself) for initial use and later reuse; writing—original draft preparation, Y.Q. and O.Z. Preparation, creation and/or presentation of the published work, specifically writing the initial draft (including substantive translation); writing—review and editing, Y.Q., O.Z. and J.M. Preparation, creation and/or presentation of the published work by those from the original research group, specifically critical review, commentary or revision—including pre-or post-publication stages; visualization, Y.Q., O.Z. and J.M. Preparation, creation and/or presentation of the published work, specifically visualization/data presentation; supervision, Y.Q. and J.M. Oversight and leadership responsibility for the research activity planning and execution, including mentorship external to the core team; project administration, Y.Q., C.L. and J.M. Management and coordination responsibility for the research activity planning and execution; funding acquisition, Y.Q. All authors have read and agreed to the published version of the manuscript

**Funding:** This research received no external funding.

**Acknowledgments:** This work was supported by Project FORDECyT 296737 "Consorcio en Inteligencia Artificial"—México.

**Conflicts of Interest:** The authors declare no conflict of interest.

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
