# Peer review of "Algorithm to Generate Trajectories in a Robotic Arm Using an LCD Touch Screen to Help Physically Disabled People"

_electronics, doi:10.3390/electronics10020104_

Round 1

Reviewer 1 Report

The major concern from the reviewer: a lot of typos and grammatical errors exist throughout the manuscript, which makes the paper really difficult to understand.

Also, the literature review work on the trajectory generation of robotic arm using an LCD touch screen is not sufficient. The proposed algorithm lacks novelty. The experimental test is missing.  

Author Response

Dear Reviewer:

I am highly thankful for your in-depth analysis that provides useful inputs to improve this manuscript.

The changes were done addressing your comments in this updated version.

Comment

Response

The major concern from the reviewer: a lot of typos and grammatical errors exist throughout the manuscript, which makes the paper really difficult to understand.

This paper was reviewed by a native person to correct any typo or grammatical errors found.

Also, the literature review work on the trajectory generation of robotic arm using an LCD touch screen is not sufficient. The proposed algorithm lacks novelty. The experimental test is missing.

Thank you for your comment. More related works have been added.

A method to generate trajectories of a robotic arm has been proposed using an LCD Touch screen and this can happen with one compute to generate the trajectory. In a difference to other methods, for example using cubic polynomial, the method requires 4 different computes and also this method is not recommended at all due to infinite acceleration. moreover, other methods require a lot of computations to gain precision. Therefore, a new method to generate the trajectory of a robotic arm has been proposed as well as the algorithm because this method and the algorithm used do not require a lot of computation, making a fast and save trajectory to help physical disabled people. The method and the results are simulated.

Reviewer 2 Report

The article presents an approach based on the algorithm to generate trajectories of an anthropomorphic robotic arm using an LCD touch screen to help rehabilitate physically disabled people. In my opinion, the topic is interesting and quite innovative. The advantage of this article is that the authors propose an algorithm to calculate the accuracy trajectory at any time using the LCD touch screen  which allowed the calculation of short trajectories with minimal finger movements. It seems to me that this method can be improved by using modern optimization methods based on artificial intelligence. This work is well planned with a good logic of consequence. The obtained results allow the reader to understand the method used.

However, I have remarks for the authors:

  1. Is this algorithm suitable when there are more joints to the robotic arm.
  2. Some notations in the equations are not clear, e.g. (ϵ), therefore I will suggest adding nomenclature in order to facilitate reading this article.
  3. Figure 8 is poorly presented, it needs better illustration.

Author Response

Dear Reviewer:

I am highly thankful for your in-depth analysis that provides useful inputs to improve this manuscript.

The changes were done addressing your comments in this updated version.

Comment

Answer

The article presents an approach based on the algorithm to generate trajectories of an anthropomorphic robotic arm using an LCD touch screen to help rehabilitate physically disabled people. In my opinion, the topic is interesting and quite innovative. The advantage of this article is that the authors propose an algorithm to calculate the accuracy trajectory at any time using the LCD touch screen which allowed the calculation of short trajectories with minimal finger movements. It seems to me that this method can be improved by using modern optimization methods based on artificial intelligence. This work is well planned with a good logic of consequence. The obtained results allow the reader to understand the method used.

Thank you very much for your comment.

However, I have remarks for the authors:

1.     Is this algorithm suitable when there are more joints to the robotic arm?

2.     Some notations in the equations are not clear, e.g. (ϵ), therefore I will suggest adding nomenclature in order to facilitate reading this article.

3.     Figure 8 is poorly presented; it needs better illustration.

1.     Yes it is, this depends on the geometric method used to calculate the radians/degrees and find the inverse kinematic to generate the trajectory as long as the geometric method used to find the inverse kinematic can be calculated using the values of (x,y) coordinates. This is explained in the lines 265 to 270.

2.     The nomenclature has been added.

3.     The Figure 8 has been changed.

Reviewer 3 Report

The paper deals with the trajectory planning of an anthropomorphic robot for rehabilitation purpose. The paper presents an algorithm for a robotic arm made up of three revolute joints. The algorithm is presented together with interesting simulations that proves the effectiveness of this work.

The paper is well written and technically sounds. Moreover its topic is in the interest of the journal.

For these reasons I propose to accept this paper after a minor revision.

In Section 4 there are these issues:

authors declare an angle alpha to be in the range 0-180 degrees, but in equation (1) alpha takes values between -90 and 90 degrees. Then, in the subsequent lines, the value of alpha is definen between 0 and 1.

The value of xp is defined as different from 0, but it is not clear the reason for this. Moreover this sounds like the robot can not move from the negative x coordinate to positive and viceversa.

In Section 5 there are these issues:

In figure 9 and 11 the robot is depicted in singular configurations. In these conditions the robot loses degrees of freedoms, so these condition should be avoided. The author may take advantage of a generic kinematic index for industrial robot to solve this problem

Finally, Figure 8 has some problems

Author Response

Dear Reviewer:

I am highly thankful for your in-depth analysis that provides useful inputs to improve this manuscript.

The changes were done addressing your comments in this updated version.

Comment

Answer

The paper deals with the trajectory planning of an anthropomorphic robot for rehabilitation purpose. The paper presents an algorithm for a robotic arm made up of three revolute joints. The algorithm is presented together with interesting simulations that proves the effectiveness of this work.

Thank you very much for your comment.

The paper is well written and technically sounds. Moreover, its topic is in the interest of the journal. For these reasons I propose to accept this paper after a minor revision.

Thank you very much for your comment.

In Section 4 there are these issues:

Authors declare an angle alpha to be in the range 0-180 degrees, but in equation (1) alpha takes values between -90 and 90 degrees. Then, in the subsequent lines, the value of alpha is defined between 0 and 1.

The value of xp is defined as different from 0, but it is not clear the reason for this. Moreover, this sounds like the robot cannot move from the negative x coordinate to positive and viceversa.

1.     A mistake has been presented in that part mentioned by you. the angle alpha has been defined between 0 and 180 degrees. Thank you.

2.     For when xp is 0, the alpha value is 90 degrees or  radians because when xp is relative small, the arctang function approximates to 90 degrees. The explanation of this has been located in the paper. The negative x coordinate is given by the value of alpha, for example when the angle alpha is between 90 degrees and 180 degrees counterclockwise, then, the coordinate x is negative and for when the angle alpha is between 90 degrees and 180 degrees clockwise, then, x is positive. Thank you for the helpful observation.

In Section 5 there are these issues:

In figure 9 and 11 the robot is depicted in singular configurations. In these conditions the robot loses degrees of freedoms, so these conditions should be avoided. The author may take advantage of a generic kinematic index for industrial robot to solve this problem

The singularity has been changed and you can see it in figure 9. This singularity (shown in figure 9) was used to obtain the final results actually. Thank you for the appreciable observation.

Finally, Figure 8 has some problems

The figure 8 has been changed.

Reviewer 4 Report

The paper is a good one and deserves publication. The introduction in terms of increasing the aspects of the possible applications of the proposed method. This could be done also increasing the cited paper justifying the choice. I put some indications in the section “Comments to the author(s)”.

The sections are organized very well and a potential readers can understand the topic also because of its very good organization.

The contribution of this work is very good. The title has been formulated unambiguously conveying the focus of the study. The accurate interpretation of outcomes, well substantiated by the results of the analysis has been achieved by them. The presentation of the results in terms of the research objectives has been successfully made.

The author/s have been able to draw logical conclusions from the results. The  results and their respective discussion points prove the efficacy of the proposed method. Conclusions are accurate and clearly based on outcomes.

Comments to the authors:

Can a singularity occur in equation (1)?

In fact, the authors say that x_p must be different from zero and the number and then a question arrises: how did you guarantee that?

Even though a singularity does not appear it could happen that x_p is relativley small.

May that generate Problem in the context of robustness?

In Paragraph 4.5 the calculation of velocity and acceleration is proposed.

Interesting should be to discuss the possibility to estimate the velocity and acceleration. In fact, in case of presence of noise measurements some inaccurances can deteriorate this calculations. Try to address this aspect also using the suggested literature here below.

The quality of the figures must be improved!

Concerning the cited literature you can consider the following paper related to the same topic to underline the importance of the proposed topic also with respect to real applications. In particular, as already discussed, in the following thwo paper a way to estimate the velocity and the acceleration is proposed.

@Article{a12050101,
AUTHOR = {Schimmack, Manuel and Mercorelli, Paolo},
TITLE = {An Adaptive Derivative Estimator for Fault-Detection Using a Dynamic System with a Suboptimal Parameter},
JOURNAL = {Algorithms},
VOLUME = {12},
YEAR = {2019},
NUMBER = {5},
ARTICLE-NUMBER = {101},
URL = {https://www.mdpi.com/1999-4893/12/5/101},
ISSN = {1999-4893},

Inside this paper published by MDPI you can find also the following interesting contribution also related to thsi topic:

 Schimmack, M.; Mercorelli, P. A sliding mode control using an extended Kalman filter as an observer for stimulus-responsive polymer fibres as actuator. Int. J. Model. Identif. Control 2017, 27, 84–91.

Mercorelli,  P. Robust feedback linearization using an adaptive PD regulator for a sensorless control of a throttle valve. Mechatron. J. IFAC 2009, 19, 1334–1345.

Dabroom, A.; Khalil, H.K. Numerical differentiation using high-gain observers. In Proceedings of the 36th IEEEConferenceonDecisionandControl,SanDiego,CA,USA, December1997;Volume5,pp. 4790–4795.

Levant, A. Robust exact differentiation via sliding mode technique. Automatica 1998, 34, 379–384. 

Levant,A. Universalsingle-input-single-output(SISO) sliding-modecontrollerswithfinite-timeconvergence. IEEE Trans. Autom. Control 2001, 46, 1447–1451. 

Yu, X.; Xu., J. Nonlinear derivative estimator. Electron. Lett. 1996, 32, 1445–1447.

Author Response

Dear Reviewer:

I am highly thankful for your in-depth analysis that provides useful inputs to improve this manuscript.

The changes were done addressing your comments in this updated version.

Comment

Response

The paper is a good one and deserves publication. The introduction in terms of increasing the aspects of the possible applications of the proposed method. This could be done also increasing the cited paper justifying the choice. I put some indications in the section “Comments to the author(s)”.

Thank you very much for your comment.

The sections are organized very well and a potential readers can understand the topic also because of its very good organization.

Thank you very much for your comment.

The contribution of this work is very good. The title has been formulated unambiguously conveying the focus of the study. The accurate interpretation of outcomes, well substantiated by the results of the analysis has been achieved by them. The presentation of the results in terms of the research objectives has been successfully made.

Thank you very much for your comment.

The author/s have been able to draw logical conclusions from the results. The results and their respective discussion points prove the efficacy of the proposed method. Conclusions are accurate and clearly based on outcomes.

Thank you very much for your comment.

Comments to the authors:

Can a singularity occur in equation (1)?

Using the tan^-1(y/x) function, a singularity at x=0 happens. Therefore, the equation (1) has been changed in order to avoid this singularity and explain better the idea of the algorithm used. Thank you for the question.

In fact, the authors say that x_p must be different from zero and the number and then a question arises: how did you guarantee that?

The function (1) has been changed, because when the algorithm is programmed to obtain the trajectory function and for when xp is relative small or xp is 0, the angle alpha is equal to 90 degrees, this is because the function tan^-1(y/x) approximates to  for when x approximates to 0, then, alpha is equal to 90 degrees when xp is really small or 0.

Even though a singularity does not appear it could happen that x_p is relatively small.

When xp approximates to 0, then tan^-1(y/x) approximates to . Then, in figure 1, the angle alpha_p has been defined equal to 90 degrees for when xp is 0 or relatively small.

May that generate Problem in the context of robustness?

Probably yes, but, for when xp is almost 0, then tan^-1(y/x) approximates to . The problem could occur if it is needed a precise position of 90 degrees. For this reason, the angle alpha_p is 90 degrees for when xp=0. Thank you for the helpful observation

Interesting should be to discuss the possibility to estimate the velocity and acceleration. In fact, in case of presence of velocity and acceleration. In fact, in case of presence of noise measurements some inaccurances can deteriorate these calculations. Try to address this aspect also using the suggested literature here below.

The papers suggested has been addressed. Thank you for the suggestion.

The quality of the figures must be improved!

Some figures have been changed by other with better quality (Figure 2, 3, 4, 7, 8, 9, 11, 15).

Concerning the cited literature, you can consider the following paper related to the same topic to underline the importance of the proposed topic also with respect to real applications. In particular, as already discussed, in the following two paper a way to estimate the velocity and the acceleration is proposed.

Thank you very much for the suggestion, the proposed articles have been reviewed and added to the references. These papers are really interesting. The topic presented in the papers can really help to estimate the velocity and the acceleration presented in the paper, for this reason, the paper has been cited for explaining future works and in the introduction to explain the important of this topic to get better results in the trajectories using any trajectory method.

The following proposed papers were included in the paper:

14.Mercorelli, P. Robust feedback linearization using an adaptive PD regulator for a sensorless control of a throttle valve. Mechatron. J. 2009, 19, 1334–1345.

15.Schimmack, M.; Mercorelli, P. A sliding mode control using an extended Kalman filter as an observer for stimulus-responsive polymer fibres as actuator. Int. J. Model. Identif. Control, 2017, 27, 84–91.

16.Schimmack, M.; Mercorelli, P. An Adaptive Derivative Estimator for Fault-Detection Using a Dynamic System with a Suboptimal Parameter. Algorithms, 2019, 12, 1-17.

Round 2

Reviewer 1 Report

(1) Although the presentation of the manuscript was improved by proofreading, there are still some grammatical errors and unprofessional expressions. For example,

i) line 56-57 on page 2: "Finally, the polynomial coefficients have to be recalculated each time it is wanted to change the end-point."

ii) line 75 on page 2: "....is lacked....".

iii) line 73 on page 2: ".... or whatever other kinds of...."

iv) line 350 on page 11: "....it is obtained...."

etc.

(2) Line 126 one page 3: "the optimal time is 13,729". What is the meaning of optimal time here? Any unit?

(3) This paper focuses on the trajectory generation of the robotic arm by using an LCD touch screen. Are there any state-of-the-art in this field, by using the LCD touch screen? Unfortunately, the reviewer did not find sufficient literature review work included.

(4) The paper title is too long.

(5)  Please check the (1,4) entry of Tables 1-3. Also, the minus sign is not correctly, looks like hyphen. For Table 2, what is the unit of alpha? Legends for curves in Figs. 9 & 11-13 are missing. 

(6) The motivation of this work is not clear.

Author Response

Reviewer

I am highly thankful for your in-depth analysis that provides useful inputs to improve this manuscript.

  1. Although the presentation of the manuscript was improved by proofreading, there are still some grammatical errors and unprofessional expressions. For example:

      i.         Line 56-57: “Finally, the polynomial coefficients have to be recalculated each time it is wanted to change the end-point.

     ii.         Line 75 on page 2: “…is lacked…”

   iii.         Line 73 on page 2: “… or whatever other kinds of…”

    iv.         Line 350 on page 11: “… it is obtained…”

These lines were changed to:

      i.         Finally, the polynomial coefficients have to be recomputed every time that a new endpoint is assigned.

     ii.          It has been proved that controlling a robotic arm by voice commands lack high reliability

   iii.         Another problem related to robotic arms control is when a joystick or another kind of sensor is used.

    iv.         Then, the position, velocity, acceleration and Jerk (the derivative of the acceleration) is obtained and shown in Figure 6….

2. Line 126 one page 3: “the optimal time is 13,729”. What is the meaning of optimal time here? Any unit?

In this comment, this paragraph was modified to explain it in a better way:

·       Then, the trajectory of the robot's joints is planned by the quantic polynomial; however, the optimal time of this method is 13.729 seconds because a smooth motion is presented at this time, and the trajectory planning presents oscillations.

3. This paper focuses on the trajectory generation of the robotic arm by using and LCD touch screen. Are there any state-of-the-art in this field, by using the LCD touch screen? Unfortunately, the reviewer did not find sufficient literature review work included. 

In this comment the following paragraph was added with related work using and LCD touch screen.

·       Some work has proposed to use a touch screen for robotic control as an alternative solution to help people with physical disabilities, providing an intuitive interface. Makwana and Tandon [17] have proposed controlling a robotic wheelchair using a touch screen to improve a person's movements with physical disabilities. Similarly, Bularka et al. [18] present an alternative to control a robotic arm using the accelerometer of a watch and a smartphone; control is carried out using finger gestures on the touch screen or moving the phone in the air. The authors in [19] present solutions for robotic control using the potential advantages of smartphones so that people with a physical disability can interact with the environment in a friendly way.

References:

·       Makwana, S. D.; Tandon, A. G. Touch screen based wireless multifunctional wheelchair using ARM and PIC microcontroller. Proceeding of International Conference on Microelectronics, Computing and Communications, Durgapur, 2016, pp. 1-4.

·       Bularka, S.; Szabo, R.; Otesteanu, M.; Babaita, M. Robotic Arm Control with Hand Movement Gestures. Proceeding of International Conference on Telecommunications and Signal Processing, Athens, 2018, pp. 1-5,

·       Wu, L.; Alqasemi, R.; Dubey, R. Development of Smartphone-Based Human-Robot Interfaces for Individuals with Disabilities, IEEE Robot. Autom. Lett. 2020, 5, 5835-5841.

4- The paper title is too long

The paper title was reduced by:

Algorithm to generate trajectories in a robotic arm using an LCD touch screen to help physically disabled people

5- Please check the (1,4) entry of Tables 1-3. Also, the minus sign is not correctly, looks like hyphen. For table 2, what is the unit of alpha? Legends for curves in Figs 9 & 11-13 are missing.

The legends for curves in Figs 10 to 13 have been added and the Tables 1-3 have been modified.

Alpha is represented in degrees; this is stated at line 405. Then, the joint-alpha, joint-2 and joint-3 results are presented using the trajectory function in each table. To obtain the unit, the result is multiplied by , that means, the unit is radians in each joint. This is explained at the beginning of section 4.4. Distribution and trajectory function and lines 377 and 378.

6- The motivation of this work is not clear.

This work's motivation focuses on helping people who are dependent or require help from other people to control these devices. Then, the trajectory captured in these devices is slower and more complex to generate a robotic arm's trajectory, and the short trajectories cannot be calculated. For this reason, it has been proposed an LCD touch screen to generate the end-point motion of the robot because the control is intuitive. Nowadays, the people are more familiarized using an LCD touch screen due to they use smartphones, laptops, etc.; which count with an LCD touch screen to work. Then, using an LCD touch screen to control an anthropomorphic robotic arm with just one finger and using the distribution function proposed is a great option to help physically disabled people in their quotidian lives with an exact, fast and comfortable alternative to control the robotic arm. Concerning the problems mentioned above when calculating a robotic arm's trajectory, it is proposed to use a distribution function, named distribution function-(ζat_s), that allows the calculation of an exact trajectory and any parameter. For example, the velocity and acceleration can be chosen without affecting the desired position and the initial position by changing one parameter and making one calculation to generate the trajectory planning.

Round 3

Reviewer 1 Report

The paper can be accepted for publication in the current form.